# Structural basis of GTPase-mediated mitochondrial ribosome biogenesis and recycling

Hauke S. Hillen [1,2,3✉], Elena Lavdovskaia [1,2], Franziska Nadler[1], Elisa Hanitsch[1], Andreas Linden[4,5], Katherine E. Bohnsack [6], Henning Urlaub [4,5] & Ricarda Richter-Dennerlein [1,2✉]

Ribosome biogenesis requires auxiliary factors to promote folding and assembly of ribosomal proteins and RNA. Particularly, maturation of the peptidyl transferase center (PTC) is mediated by conserved GTPases, but the molecular basis is poorly understood. Here, we define the mechanism of GTPase-driven maturation of the human mitochondrial large ribosomal subunit (mtLSU) using endogenous complex purification, in vitro reconstitution and cryo-EM. Structures of transient native mtLSU assembly intermediates that accumulate in GTPBP6-deficient cells reveal how the biogenesis factors GTPBP5, MTERF4 and NSUN4 facilitate PTC folding. Addition of recombinant GTPBP6 reconstitutes late mtLSU biogenesis in vitro and shows that GTPBP6 triggers a molecular switch and progression to a near-mature PTC state. Additionally, cryo-EM analysis of GTPBP6-treated mature mitochondrial ribosomes reveals the structural basis for the dual-role of GTPBP6 in ribosome biogenesis and recycling. Together, these results provide a framework for understanding step-wise PTC folding as a critical conserved quality control checkpoint.

[1] Department of Cellular Biochemistry, University Medical Center Goettingen, Goettingen, Germany. [2] Cluster of Excellence "Multiscale Bioimaging: from Molecular Machines to Networks of Excitable Cells" (MBExC), University of Goettingen, Goettingen, Germany. [3] Research Group Structure and Function of Molecular Machines, Max Planck Institute for Biophysical Chemistry, Goettingen, Germany. [4] Bioanalytical Mass Spectrometry Group, Max Planck Institute for Biophysical Chemistry, Goettingen, Germany. [5] Bioanalytics, Institute for Clinical Chemistry, University Medical Center Goettingen, Goettingen, Germany. [6] Department of Molecular Biology, University Medical Center Goettingen, Goettingen, Germany. ✉email: hauke.hillen@med.uni-goettingen.de; ricarda.richter@med.uni-goettingen.de

The human mitochondrial ribosome (mitoribosome) synthesizes 13 essential subunits of the oxidative phosphorylation (OXPHOS) system. Defects in mitoribosome biogenesis or function result in OXPHOS deficiency and cause severe early-onset mitochondrial diseases[1]. The mitoribosome is composed of a small and a large ribosomal subunit (mtSSU and mtLSU, respectively), which each contain a ribosomal RNA (12S, 16S rRNA) and a number of mitoribosomal proteins (MRPs). Structural analyses of the mitoribosome have revealed its overall architecture and differences to cytoplasmic and bacterial ribosomes[2,3]. However, little is known about the mechanisms of mitoribosome assembly.

Biogenesis of the large ribosomal subunit (LSU) in all translation systems proceeds through distinct steps in which the formation of the peptidyl transferase center (PTC), the active site of the ribosome, represents the last and most critical step and requires the assistance of assembly factors[4–7]. In particular, universally conserved GTPases, which act as quality control and antiassociation factors, play key roles during late LSU assembly stages and ensure proper maturation of the PTC[8]. In human mitochondria, the GTPases GTPBP5, GTPBP6, GTPBP7, and GTPBP10 mediate mtLSU maturation, as their deletion results in a loss of active mitoribosomes and stalls mtLSU biogenesis at distinct states[8–14]. In particular, ablation of either GTPBP5 or GTPBP6 in human cells results in the accumulation of late mtLSU assembly intermediates containing GTPBP7, GTPBP10, MALSU1, MTERF4, NSUN4 and, in the case of GTPBP6 loss, GTPBP5[9,12,14]. MTERF4 is a putative RNA-binding protein that forms a stable complex with the methyltransferase NSUN4[15,16], which modifies the 12S rRNA during mtSSU biogenesis[17]. However, the role of the MTERF4-NSUN4 complex in mtLSU assembly is unclear. Previous structural studies of mtLSU assembly intermediates purified from wild-type human cells revealed premature rRNA conformations and showed that MALSU1 forms a submodule with L0R8F8 and mtACP that binds to the mtLSU and likely prevents premature subunit association[5]. These results indicate that the GTPases act hierarchically and cooperate with other maturation factors during mtLSU biogenesis. However, the mechanistic basis of GTPase-driven LSU biogenesis and PTC folding is not known.

Here, we combine genetic perturbation and endogenous complex purification with in vitro reconstitution and cryo-electron microscopy (cryo-EM) to dissect the molecular basis of GTPase-mediated human mtLSU maturation and mitoribosome recycling.

## Results and discussion

### Two distinct mtLSU biogenesis intermediates accumulate in the absence of GTPBP6.
We isolated mtLSU complexes from a human cell line lacking GTPBP6, which accumulate assembly intermediates that contain GTPBP7, GTPBP10, and GTPBP5 as well as MALSU1, MTERF4, and NSUN4[12]. Mass-spectrometric analyses confirmed the presence of these proteins as well as all 52 MRPs (Supplementary Fig. 1a, Supplementary Data 1). We then analyzed the intermediates by single-particle cryo-EM (dataset 1). Particle classification yielded two distinct reconstructions of mtLSU assembly intermediates at overall resolutions of 2.2 and 2.5 Å, respectively, which led to refined structures (Supplementary Fig. 1b–e, Supplementary Table 1).

Compared to the mature mtLSU, both reconstructions show an extra density close to the L1 stalk and a distinct folding of the interfacial rRNA (Supplementary Fig. 1c). The density could be unambiguously fit with the crystal structure of the complex of the RNA-binding protein MTERF4 and the methyltransferase NSUN4[15,16], with some adjustments (Supplementary Fig. 2a, b).

Additionally, both structures contain all MRPs as well as the MALSU1-L0R8F8-mtACP module, which prevents premature subunit association[5]. However, they differ in their PTC conformation, and the second reconstruction showed an additional density above the PTC. Based on its resemblance to bacterial Obg[18], we identified it as GTPBP5 (Supplementary Fig. 2c). Thus, two distinct mtLSU biogenesis intermediates accumulate in the absence of GTPBP6, one containing MTERF4-NSUN4 and one additionally containing GTPBP5.

**MTERF4 binds subunit-bridging elements in the rRNA.** The structure of the MTERF4-NSUN4-bound assembly intermediate reveals a distinct rRNA conformation, which appears to be stabilized by binding of MTERF4-NSUN4 to the interfacial side of the 16S rRNA (Fig. 1a, b). MTERF4 interacts with the mtLSU through contacts with h75 in the L1 stalk (nucleotides 2743–2756 and 2792–2804) and with the C-terminal tail of uL2m (residues 275–300) (Fig. 1c). In the mature mtLSU, this tail runs in between h68 and h66[3]. In the MTERF4-bound state, it is rearranged beneath MTERF4 and forms a helix (residues 290–295) that binds α12 of MTERF4. Compared to previous crystal structures, the curvature of MTERF4 is altered by an inward rotation of helices α1–8, resulting in a narrower RNA-binding groove (Supplementary Fig. 2b). NSUN4 binds to the mtLSU on top of the P loop/h80 (nucleotides 2815–2821) and h81 (nucleotides 2841–2853) (Fig. 1c), next to bL33m. The N-terminal part of mL64 runs above NSUN4, but this region is invisible, as in previous structures[3]. The active site of NSUN4 is positioned above h81, but the closest RNA base is more than 15 Å away from its SAM-binding site[15,16], suggesting that NSUN4 does not methylate the 16S rRNA, as had been shown previously[19].

MTERF4 additionally contacts the interfacial rRNA segment that forms h68–h70 in the mature mtLSU. In the MTERF4-NSUN4 assembly intermediate, this region is partially unfolded and wraps over the outward-facing RNA-binding groove of MTERF4 (Fig. 1d). Although the density did not allow for atomic modeling, comparison shows that helices h68 and h69 would clash with MTERF4 in their mature conformation (Fig. 1d), indicating that MTERF4 binds a premature conformation of the interfacial rRNA. In the mature mitoribosome, h68–h70 form seven of the fifteen intersubunit bridges[2]. MTERF4-NSUN4 may thus act as a quality-control checkpoint by sequestering the interfacial rRNA to prevent subunit joining prior to final mtLSU maturation.

**GTPBP5 and NSUN4 cooperate to facilitate PTC folding.** The structure of the second mtLSU intermediate reveals the structure and function of the Obg domain of GTPBP5 (Fig. 2a, b). GTPBP5 binds above the PTC and interacts primarily with the rRNA (Supplementary Fig. 2d). It consists of an N-terminal Obg domain and a C-terminal GTPase domain (Fig. 2b). The GTPase domain is poorly resolved, but the density indicates that it resides between uL11m and the sarcin-ricin loop (SRL, h95). The Obg domain extends along the PTC and contains three conserved loops at its tips (loop 1: residues 93–103, loop 2: residues 140–145, loop 3: residues 191–205)[18] (Fig. 2b, Supplementary Fig. 2c, d). These loops reach into the PTC, where we additionally observe density for the N-terminal tail of NSUN4 (residues 26–37), which was invisible in the intermediate lacking GTPBP5 but becomes ordered in the presence of GTPBP5 (Fig. 2c, d).

Comparison of the two assembly intermediates shows that they differ in their PTC maturation state. In the intermediate lacking GTPBP5, the PTC is partially disordered and adopts a premature conformation (Fig. 2c). In particular, h72 (2692–2695), the PTC loop (nucleotides 2936–2946 and 2977–2992), h39 (nucleotides

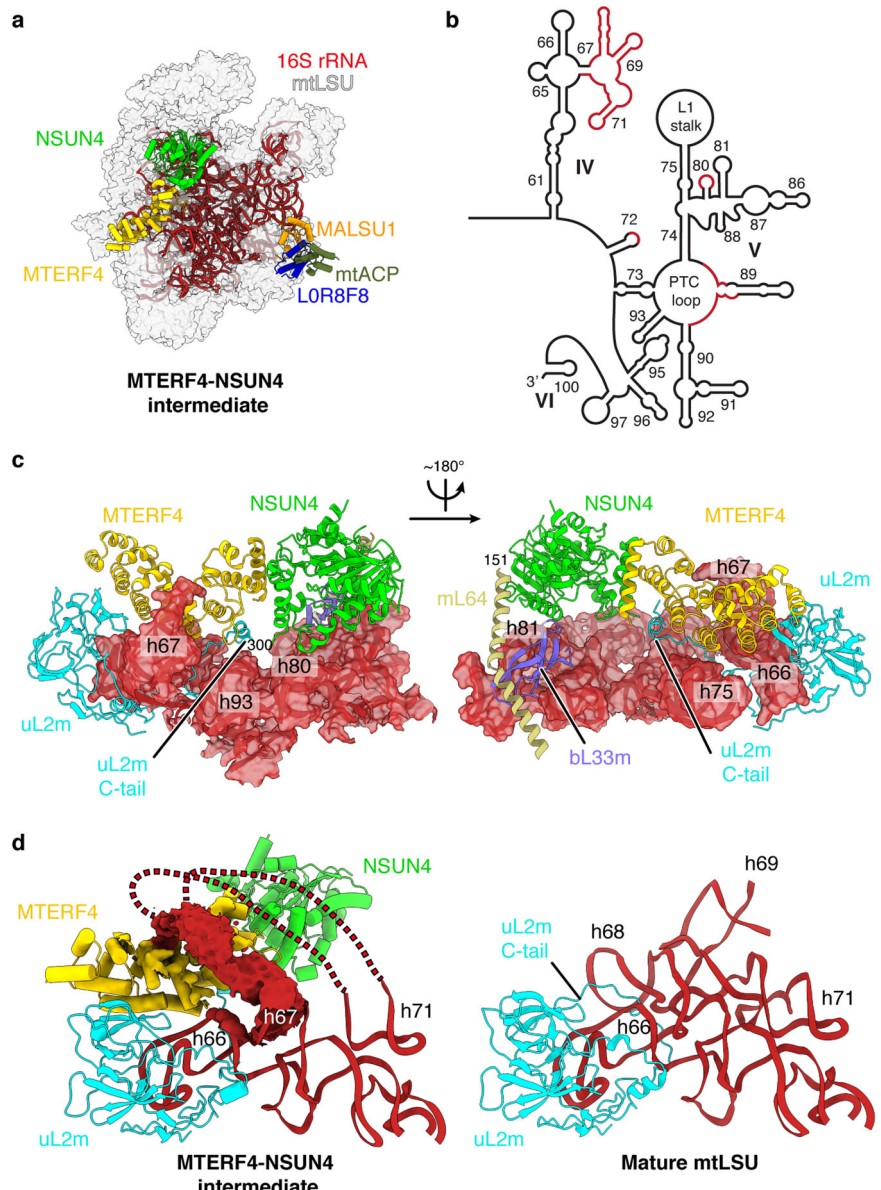

**Fig. 1 Structure of the MTERF4-NSUN4-bound mtLSU assembly intermediate. a** Cryo-EM structure of the MTERF4-NSUN4-bound large mitoribosomal subunit (mtLSU) intermediate (dataset 1). The 16S rRNA (red) and indicated biogenesis factors are shown as cartoon and the remaining mitoribosomal proteins (MRPs) are shown as gray transparent surface. NSUN4: lime green, MTERF4: yellow, MALSU1: orange, mtACP: olive, L0R8F8: blue. **b** Schematic depiction of the mature 16S rRNA secondary structure with domains IV–VI as indicated. Regions with distinct fold in the MTERF4-NSUN4 intermediate are depicted in red. **c** Interaction of the MTERF4-NSUN4 complex with the mtLSU. Regions of the 16S rRNA interacting with MTERF4-NSUN4 are shown as cartoon and as transparent surface. MTERF4, NSUN4, uL2m, bL33m, and mL64 are shown in cartoon representation. Coloring as in **a** and as follows: uL2m: cyan, bL33m: violet, mL64: sand. **d** (Left) In the MTERF4-NSUN4 assembly intermediate, the 16S rRNA region encompassing h68–h71 wraps above MTERF4, as indicated by red lines. The density is shown as red surface. (Right) Closeup view of the 16S rRNA region 2469–2659 in the mature mtLSU (PDB 3J7Y)[3]. The fold observed in the mature mtLSU would clash with MTERF4-NSUN4.

2108–2115), and the P loop/h80 (nucleotides 2814–2818) adopt distinct conformations. The PTC loop is partially mobile, and nucleotides 2975–2995 form a loop that base pairs with the tip of the P loop/h80 (Fig. 2c). In the GTPBP5-containing intermediate, the PTC adopts a more mature-like conformation in which the PTC loop forms the base of h89, and the h80 tip is shifted upward (Fig. 2d). The structures show how GTPBP5 and the NSUN4 tail facilitate these rearrangements. First, GTPBP5 and the NSUN4 tail disrupt the interaction between the PTC loop and h80, which allows the PTC loop to refold. NSUN4 sequesters bases G2817 and G2814 through stacking interactions with W31 and Y27, respectively (Fig. 2d), and GTPBP5 binds G2816 through stacking

interactions with F92 and sandwiches A3089 between F100, A202 and F92. Second, GTPBP5 stabilizes the refolded PTC conformation through backbone and base interactions with charged residues in loop 1 and loop 3 (R97, K98, E95, E99, and R201) (Fig. 2d). These rearrangements also lead to ordering of ribosomal protein elements, which are disordered in the intermediate without GTPBP5, including parts of mL63 (residues 9–21), uL10m (residues 30–36), and uL16m (residues 47–69 and 134–148).

PTC maturation also involves 2'-O-methylation of bases within the P loop/h80 (G2815) and the A loop/h92 (U3039 and G3040) catalyzed by MRM1, MRM2 and MRM3, respectively[14,20,21].

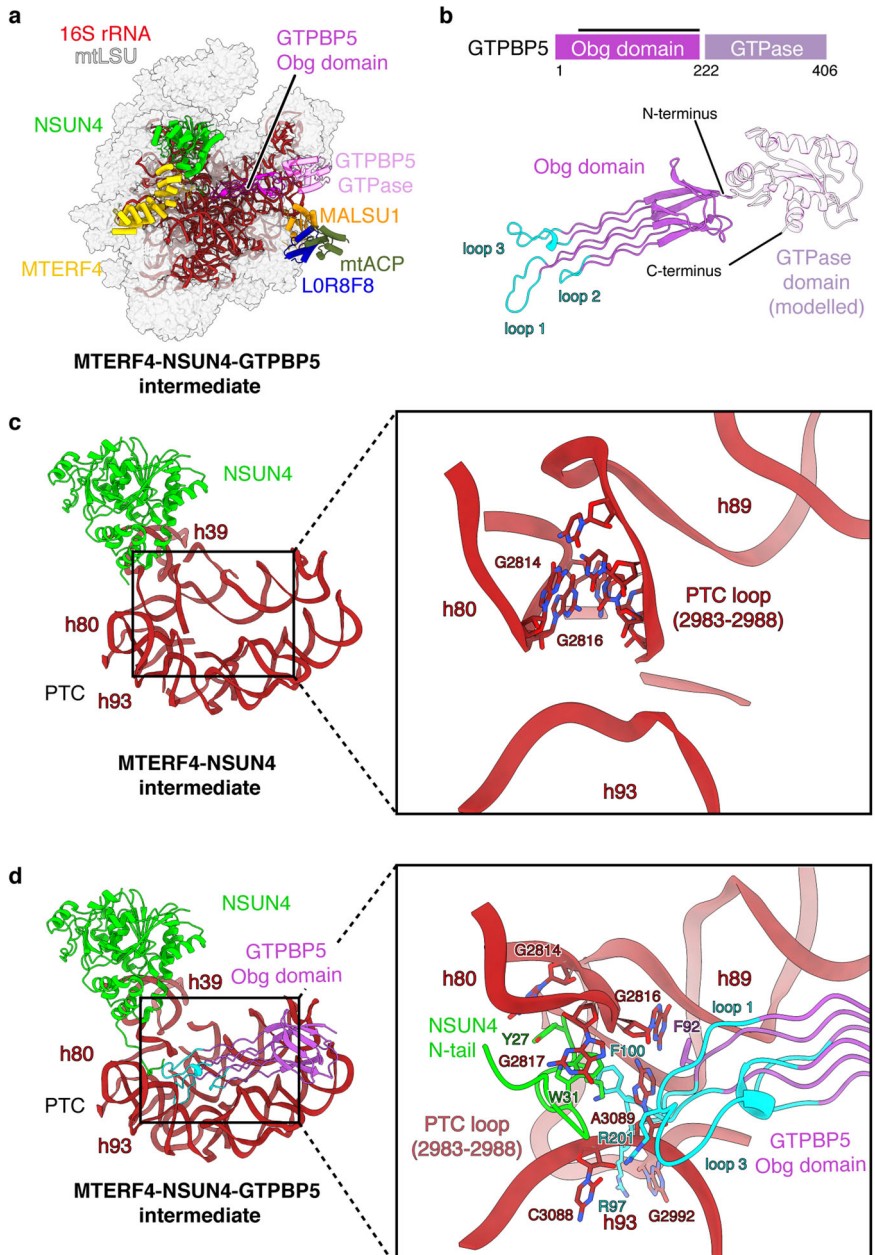

**Fig. 2 GTPBP5 cooperates with NSUN4 to mature the PTC. a** Cryo-EM structure of MTERF4-NSUN4-GTPBP5-bound large mitoribosomal subunit (mtLSU) intermediate (dataset 1). Depiction as in Fig. 1a with GTPBP5 shown in pink. **b** Structure of human GTPBP5. The protein is depicted schematically on top, with domain annotation indicated by coloring and residue numbers. The black bar above represents regions modeled in the structure. A homology model of the GTPase domain is shown transparently, but was not included in the final model. The conserved loops 1–3 that contact bases of the peptidyl transferase center (PTC) are shown in cyan. **c**, **d** The P loop/h80, h93, and PTC loop undergo rearrangements upon GTPBP5 binding. Closeup view of the PTC in the NSUN4-MTERF4 (**c**) and NSUN4-MTERF4-GTPBP5 (**d**) mtLSU assembly intermediates. The rRNA and proteins are shown as cartoons, with coloring as in Fig. 1. Selected bases that undergo rearrangements are indicated and highlighted as sticks. Loop 1 and 3 of GTPBP5 and the N-terminal tail of NSUN4 are shown as cartoons and residues that interact with the indicated bases are shown as sticks.

Methylation of U3039 is one of the last steps in PTC maturation and requires GTPBP5[14]. The cryo-EM reconstructions show density consistent with 2'-O-methylation at all these residues (Supplementary Fig. 2e), suggesting that GTPBP6 acts downstream of MRM1-3 and GTPBP5, in agreement with previous data[12]. In a complementary study, Cipullo et al.[22] report structures of mtLSU intermediates in complex with GTPBP5 and MRM2, which lack density for 2'-O-methylated U3039, thus indicating that these intermediates represent assembly states upstream of the MTERF4-NSUN4-GTPBP5 intermediate described here. This suggests that MRM2 may be independently released upon catalysis while GTPBP5 can remain bound, at least in the absence of GTPBP6.

Taken together, the cryo-EM structures reveal how GTPBP5 and NSUN4 facilitate maturation of the PTC and suggest sequential PTC methylation and folding (Supplementary Movie 1). They also explain the dual-role of NSUN4 as a methyltransferase in mtSSU assembly[17] and as a biogenesis factor

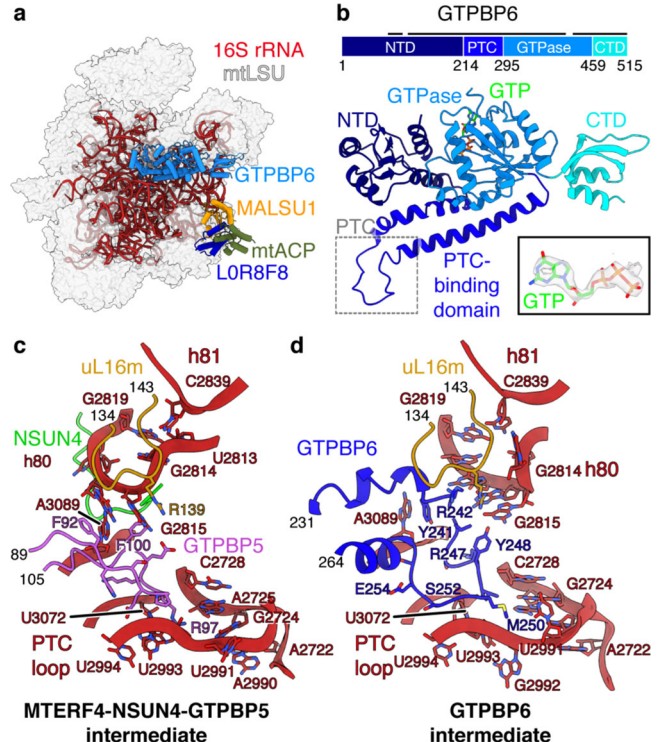

**Fig. 3 GTPBP6 binding causes rearrangements in the PTC. a** Cryo-EM structure of the GTPBP6-bound large mitoribosomal subunit (mtLSU) assembly intermediate (dataset 2). Depiction as in Fig. 1a with GTPBP6 in marine. **b** Structure of human GTPBP6. The protein is depicted schematically on top, with domain annotation indicated by coloring and residue numbers. The black bar above represents regions modeled in the structure. The density for the bound GTP is shown as indent. **c, d** Rearrangements in the peptidyl transferase center (PTC) upon GTPBP6 binding. The PTC region is shown enlarged in the MTERF4-NSUN4-GTPBP5 (**c**) and the GTPBP6 (**d**) mtLSU assembly intermediates. The rRNA and proteins are shown as cartoons, with coloring as in Fig. 2 and uL16m in ocher. NTD N-terminal domain, CTD C-terminal domain.

in mtLSU maturation, because the latter function is mediated by its N-terminal tail, which is not conserved in homologous methyltransferases that act only as small-subunit biogenesis factors, such as bacterial rsmB.

**GTPBP6 displaces GTPBP5 and MTERF4-NSUN4.** We next aimed to investigate the role of GTPBP6 during mtLSU biogenesis. For this, we reconstituted mtLSU maturation in vitro by complementing the biogenesis intermediates from GTPBP6-deficient cells with recombinant GTPBP6 in the presence of GTP and ATP. Subsequent cryo-EM analysis again revealed the MTERF4-NSUN4- and the MTERF4-NSUN4-GTPBP5-bound mtLSU complexes (dataset 2; Supplementary Fig. 3, Supplementary Table 2), which were largely identical as before but showed improved density for the interfacial rRNA and the GTPBP5 GTPase domain (see experimental procedures) (Supplementary Fig. 4a–d). In addition, classification yielded a 2.6 Å reconstruction that lacks MTERF4-NSUN4 and GTPBP5 but shows a new density that corresponds to GTPBP6 in close proximity to the L12 stalk, which allowed us to build a molecular model (Fig. 3a, b, Supplementary Figs. 3, 4e). GTPBP6 contains a N-terminal nucleotide-binding domain (NTD), a PTC-binding linker domain, a GTPase domain and a C-terminal domain (CTD) (Fig. 3b). Like GTPBP5, it binds above the PTC and primarily

interacts with rRNA (Supplementary Fig. 4f). The NTD stacks against h71, which appears stabilized compared to the MTERF4-NSUN4 state. By analogy with its bacterial homolog HflX[12], the NTD of GTPBP6 is predicted to be a putative ATP-dependent RNA helicase domain. However, we did not observe density for a bound ATP molecule, suggesting that ATP is not required for its function during ribosome biogenesis. The PTC-binding domain resides between h89 and h92, and inserts a loop (residues 241–255) deep into the PTC. The GTPase domain is located next to the NTD, on top of h89, and shows clear density for a bound GTP molecule, suggesting that GTP hydrolysis is not required for GTPBP6 binding (Fig. 3b, Supplementary Fig. 4g). The CTD is positioned between the SRL (h95) and uL11m, in a similar position as the GTPase domain of GTPBP5 (Supplementary Fig. 4f, h).

Comparison of the GTPBP6- and GTPBP5-bound structures suggests a mechanism for the hierarchical action of these factors during mtLSU biogenesis. Superimposition shows that both factors share the same binding site, and that the NTD of GTPBP6 would clash with NSUN4 (Supplementary Fig. 4h). Thus, binding of GTPBP6 and GTPBP5/MTERF4-NSUN4 is mutually exclusive, suggesting that these factors act sequentially and that GTPBP6 either triggers the release of GTPBP5/MTERF4-NSUN4 from the mtLSU or binds after their dissociation.

**GTPBP6 mediates PTC maturation.** Comparison to the GTPBP5-bound state also reveals that GTPBP6 further folds the PTC. The PTC-binding loop of GTPBP6 occupies the same position as loop 1 of GTPBP5, between the P loop/h80 and the PTC loop and in vicinity to uL16m (residues 134–143), and causes rearrangements in the elements (Fig. 3c, d, Supplementary Movie 1). In particular, the tip of the P loop/h80 (residues 2814–2819) refolds, which leads to elimination of a basepair between G2819 and U2813 and replacement of the latter by C2839 from h81. G2814 is flipped out of the loop to face outward, and G2815 is rearranged and may contact Y248 in GTPBP6 and R139 in uL16m. Binding of GTPBP6 also leads to movement of A3089 by replacing its interaction with F92 in GTPBP5 by a stacking interaction with Y241 in GTPBP6. Finally, nucleotides 2990–2994, 2722–2728 and U3072 in the PTC loop undergo conformational changes, which may be induced by a change in chemical environment upon exchange of maturation factors, as GTPBP6 places hydrophobic residues (L249, M250) at the position previously occupied by R97 of GTPBP5. Collectively, these rearrangements lead to a nearly mature PTC conformation.

**A molecular model of GTPase-mediated PTC maturation.** These structural snapshots allow us to deduce a model for GTPase-mediated mtLSU assembly and PTC maturation, which we have summarized in a molecular movie (Fig. 4, Supplementary Movie 1). First, early biogenesis factors assemble a core mtLSU with a premature PTC and partially unfolded interfacial rRNA that lacks late-stage MRPs as well as 2'-O-methylations within the PTC[14]. This mtLSU intermediate contains the MALSU1-L0R8F8-mtACP module and the MTERF4-NSUN4 complex, which both prevent premature subunit joining[5]. PTC methylations are then introduced by MRM1 and MRM2, leading to the intermediate with methylated but unfolded PTC. Next, GTPBP5 and NSUN4 act in concert to establish the basic architecture of the PTC. Both GTPBP5 and the MTERF4-NSUN4 are then replaced by GTPBP6, which induces further conformational changes that lead to a near-mature PTC. The release of MTERF4-NSUN4 liberates the rRNA region h68–h71, which can then adopt its final conformation to facilitate subunit joining. Final maturation steps must then involve dissociation of GTPBP6 and the MALSU1-

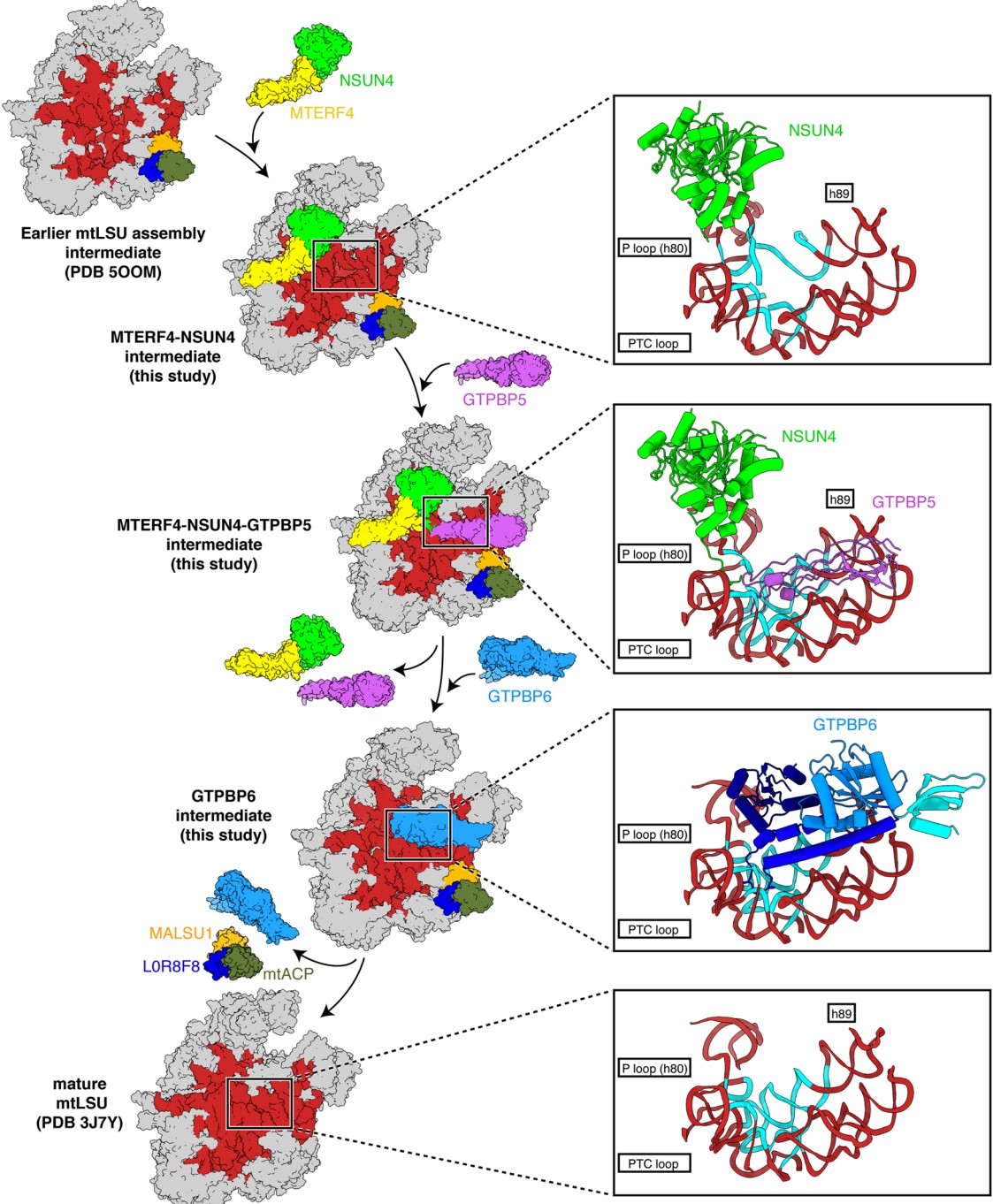

**Fig. 4 Model of GTPase-mediated maturation of human mtLSU.** Intermediate states of the mtLSU are depicted as surface with coloring as in Figs. 1–3. Models of immature and mature mtLSU were derived from previous studies (PDB 5OOM, 3J7Y)[3,5]. Closeup views illustrate GTPase-mediated rearrangements in the peptidyl transferase center (PTC). PTC intermediate states are depicted as in Fig. 2 and Supplementary Fig. 4, with the regions undergoing conformational changes highlighted in cyan.

L0R8F8-mtACP module to allow formation of the functional 55S mitoribosome.

In addition to GTPBP5, we also observed the accumulation of GTPBP7 and GTPBP10 in the mtLSU samples isolated from GTPBP6-deficient cells (Supplementary Data 1). However, despite extensive classification efforts, we did not identify particle populations with these factors present in our cryo-EM datasets. During the revision of our manuscript, several complementary studies were published showing the binding site of these GTPases on the mtLSU[22–24]. Like GTPBP5, GTPBP10 is a homolog of bacterial ObgE and accommodates

the same position on the mtLSU, which suggests that binding of GTPBP5 and GTPBP10 is mutually exclusive[23]. Previous studies indicate that GTPBP10 is among the first GTPases that binds to the mtLSU to facilitate the final maturation steps, and thus likely acts before GTPBP5[8,11,13]. This could explain the absence of GTPBP10-containing assembly intermediates in our datasets, as assembly of the mtLSU may progress past the GTPBP10-bound state even in GTPBP6-deficient cells. GTPBP7 appears to be somewhat flexible and may bind to the mtLSU in different orientations[22,24,25]. The absence of detectable GTPBP7-containing particles in our mtLSU intermediates purified from

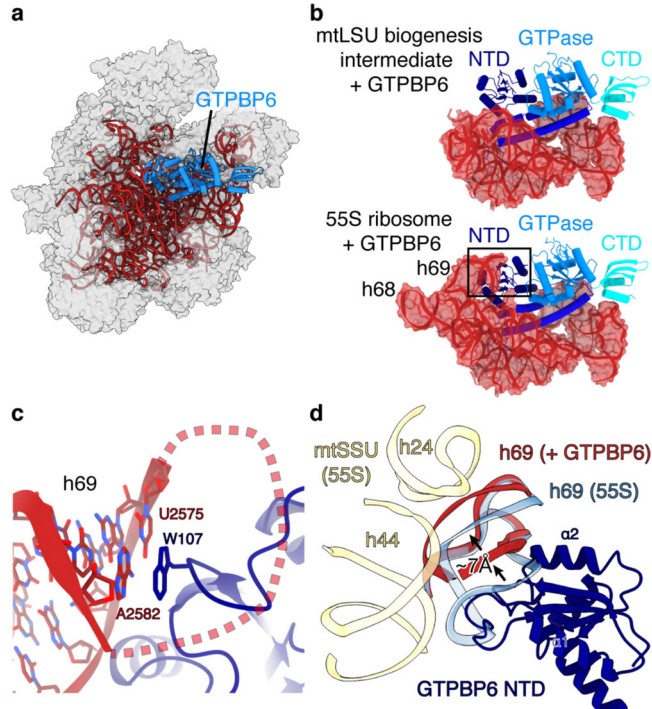

**Fig. 5 Structural basis of ribosome recycling by GTPBP6. a** Cryo-EM structure of GTPBP6 bound to the mature large mitoribosomal subunit (mtLSU) (dataset 3). The 16 s rRNA and GTPBP6 are shown as cartoon and colored as in Fig. 3. **b** Comparison of 16 rRNA interactions of GTPBP6 during biogenesis (top) and ribosome splitting (bottom). **c** GTPBP6 interacts with h69. Closeup view of h69 in the split mature mtLSU bound to GTPBP6. W105, which forms stacking interactions with h69, is shown as sticks. **d** Comparison of intersubunit-bridging elements in the 55S mitoribosome and the GTPBP6-bound split mtLSU. The GTPBP6-bound split mtLSU structure was superimposed with the structure of the elongating 55S ribosome (PDB 6ZSG)[52]. h69 is shown in blue (55S mitoribosome) or red (GTPBP6-bound mtLSU). GTPBP6 binding causes a shift of h69 by ~7 Å leading to clashes with h24 and h44 (yellow) in the small mitoribosomal subunit (mtSSU). NTD: N-terminal domain, CTD C-terminal domain.

GTPBP6-deficient cells indicates that under these conditions, GTPBP7-binding may be highly transient.

**Mechanism of GTPBP6-mediated ribosome recycling.** In addition to its role in mtLSU biogenesis, GTPBP6 also facilitates the dissociation of intact 55S mitoribosomes into subunits, which might be required to rescue stalled ribosomes[12]. To determine the mechanism of GTPBP6-mediated ribosome splitting, we treated 55S mitoribosomes with recombinant GTPBP6 in the presence of GTP and ATP. Subsequent cryo-EM analysis revealed the presence of complete mitoribosomes as well as free mtLSU and mtSSU particles (Supplementary Fig. 5, Supplementary Table 3). Classification of the mtLSU particle population led to a reconstruction at an overall resolution of 2.7 Å, which shows that GTPBP6 binds to the mature mtLSU in the same location as during ribosome biogenesis (Fig. 5a, b). As before, we observe GTP bound to the GTPase domain but no ATP in the NTD. This is in agreement to previous data, which showed that GTPBP6-mediated ribosome recycling requires GTP, but no ATP[12]. In addition to the previously observed contacts, GTPBP6 also interacts with h69 in the mature mtLSU (Fig. 5b, c), which was unfolded in the biogenesis intermediates but forms intersubunit

contacts with the mtSSU in the 55S mitoribosome. The NTD of GTPBP6 binds to h69 and inserts a tryptophan residue (W107) next to the U2575-A2582 basepair at its tip. Superimposition with the intact mitoribosome shows that h69 is shifted by ~7 Å in the GTPBP6-bound state, which would lead to clashes with h44 in the mtSSU (Fig. 5d). Thus, GTPBP6 dissociates the ribosome by rearranging elements that mediate intersubunit interactions, as has been suggested for HflX and ribosome recycling factors[26–29]. However, its mechanism is distinct, because W107 is not conserved and HflX does not form direct contacts with bases in h69, yet causes a more prominent displacement (13 vs 7 Å)[28].

The structure of GTPBP6 bound to the mature mtLSU also reveals two distinct conformations of the PTC-binding loop (Supplementary Fig. 6a). While the first is similar to that observed during mtLSU biogenesis, the second shows rearrangements in α7 and α8 and a register shift in α5 that translates into the PTC. This is accompanied by rearrangements of PTC bases, including A3089 (E. coli: A2602), U3072 (E. coli: 2585), and U2993 (E. coli: U2993) (Supplementary Fig. 6b), which play key roles during peptide bond formation and peptide release[30–34] and undergo conformational changes during elongation and termination in bacteria[32,33]. However, it is not clear whether GTPBP6 induces these rearrangements or recognizes different PTC configurations that occur during the peptide elongation cycle.

The preferred substrates of GTPBP6 are vacant ribosomes or posthydrolysis complexes with a deacylated tRNA in the P site[12]. Our structural data explain this preference, as a peptidyl tRNA in the P site would prevent GTPBP6 binding. To recycle ribosomes, release factors that trigger peptide hydrolysis such as ICT1/mL62 must therefore act prior to GTPBP6, as has been suggested recently[12,28,35]. Alternatively, GTPBP6-mediated ribosome recycling may require spontaneous hybrid P/E state formation, in agreement with in vitro data[12].

In summary, these results provide the structural and mechanistic basis of late mtLSU maturation by showing how universally conserved and mitochondria-specific assembly factors act in concert to mediate the step-wise folding of the PTC. This provides an important step towards understanding ribosome biogenesis in general, as GTPase-driven PTC maturation appears to be a conserved quality-control step during ribosome biogenesis throughout different domains of life. This is underscored by the high degree of structural similarity between bacterial and human mitochondrial ribosome biogenesis factors. However, their mechanisms appear to have at least partially diverged. Recent structural studies suggest that the bacterial GTPBP5/GTPBP10 homolog ObgE cooperates with the MALSU1-homolog RsfS and with factors that are not conserved in human mitochondria to prevent premature subunit joining and facilitate folding of h89[36]. In contrast, GTPBP5 acts in concert with the mitochondria-specific N-terminal tail of NSUN4, which in turn binds MTERF4 that prevents interfacial rRNA maturation. Thus, human mitochondria appear to combine conserved ribosome maturation factor folds with organelle-specific mechanisms to achieve mtLSU maturation.

Our findings are supported by several recently published complementary studies, which provide detailed structural snapshots of earlier mtLSU maturation steps mediated by GTPBP7, GTPBP10 and GTPBP5 in conjunction with MRM2[22–25]. Taken together, a comprehensive mechanistic picture of late mtLSU biogenesis now emerges from this large body of structural and functional information.

## Methods
No statistical methods were used to predetermine sample size. The experiments were not randomized, and the investigators were not blinded to allocation during experiments and outcome assessment.

**Cell culture conditions**. HEK293-Flp-In T-Rex wild-type (WT) (Thermo Fisher Scientific) and *Gtpbp6*[−/−] cell lines were grown under standard cultivation conditions[12]. Briefly, cells were cultured in Dulbecco's modified Eagle's medium (DMEM) with supplements (10% FBS (Sigma), 2 mM L-glutamine (GIBCO), 1 mM sodium pyruvate (GIBCO), 50 µg/ml uridine (Sigma–Aldrich)) in 5% $CO_2$ humidified atmosphere at 37 °C.

**Mitoplasts isolation**. Cells from 100–120 cell culture plates (15 cm) were harvested and homogenized in trehalose buffer (300 mM trehalose, 10 mM KCl, 10 mM HEPES-KOH pH 7.4) with addition of 1 mM PMSF and 0.2% BSA using Homogenplus Homogenizer (Schuett-Biotec, Germany). After each homogenization step mitochondria were separated from cell debris and nuclei at $1000 \times g$ for 10 min, 4 °C. Obtained mitochondria were pelleted for large-scale mitoplast preparation. To isolate mitoplasts, mitochondria were subjected to digitonin/proteinase K treatment (detergent to protein ratio 1:4, proteinase K to protein ratio 1:200), washed 4 times with trehalose buffer and pelleted at $25,000 \times g$ for 15 min, 4 °C (SS34 Rotor, Beckman Coulter).

**Purification of mitoribosomes**. Mitoplasts were lysed in lysis buffer (20 mM Tris-HCl (pH 7.4), 100 mM $NH_4Cl$, 15 mM $MgCl_2$, 2 mM DTT, 1% Triton X-100) with detergent to protein ratio 2.5:1 and the resulting lysate was clarified by centrifugation at $16,000 \times g$ for 15 min, 4 °C. To enrich mitoribosomal particles and to eliminate other mitochondrial protein complexes contaminants, the collected supernatant was subjected to a two-step sucrose cushion (1 M sucrose cushion/ 1.75 M sucrose cushion) and centrifuged for 15 h at $148,000 \times g$ at 4 °C. Fractions were collected from top to bottom of the cushion and the fraction containing mitoribosomal particles was concentrated and subsequently washed with 5 volumes of wash buffer (100 mM $NH_4Cl$, 15 mM $MgCl_2$, 20 mM Tris-HCl (pH 7.4), 2 mM DTT) in order to reduce the sucrose concentration and the sample volume. Concentrated sample was loaded on a 15–30% sucrose gradient (15–30% (w/v) sucrose, 100 mM $NH_4Cl$, 15 mM $MgCl_2$, 20 mM Tris-HCl (pH 7.4)), centrifuged at $115,600 \times g$ for 16 h 10 min, 4 °C (SW41Ti rotor, Beckman Coulter) and 16 fractions were collected. Each fraction was measured at 260 nm and fractions corresponding to the mtLSU or 55S mitoribosome were further concentrated and washed with wash buffer as described above. Purified mitoribosomal complexes were analysed by western blotting and stored at −80 °C, or used directly for grid preparation or for in vitro reconstitutions followed by cryo-EM analyses.

To monitor the structural rearrangements in the mtLSU upon GTPBP6-binding 0.1–0.18 µM of 39 s/55 s particles purified from *Gtpbp6*[−/−] or WT cell lines were mixed with 20-fold molar excess of purified GTPBP6 protein in the presence of nucleotides (1 mM ATP/1 mM GTP) in reaction buffer (2 mM DTT, 100 mM $NH_4Cl$, 15 mM $MgCl_2$, 20 mM Tris-HCl (pH 7.4)). Mixtures were incubated at 4 °C for 30 min and were used directly for grid preparation.

**Western blotting and immunodetection**. After separation of the mitoribosomal complexes by sucrose gradient ultracentrifugation, 16 fractions were collected and analyzed by western blotting. For each fraction, ~1.4% of the total volume was resuspended in loading buffer containing 2% (w/v) SDS and 50 mM DTT (final concentration) and loaded on a 10–18% SDS Tris-Tricine gel. Ten micrograms of crude mitochondria isolated from HEK293T WT cells were used as a control. Proteins were transferred to nitrocellulose membrane Amersham™ Protran™ 0.2 µM NC (GE Healthcare) and visualized using specific antibodies. Primary polyclonal anti-rabbit antibodies used in this study (dilution is indicated in brackets): anti-NSUN4 [1:1000] (ref. [12] ProteinTech; #16320-1-AP), anti-MTERF4 [1:1000] (refs. [12,14] Sigma Prestige; #HPA027097; RRID:AB_10603879), anti-MALSU1 [1:1000] (ref. [11] Proteintech; #22838-1-AP; RRID:AB_11182483), anti-GTPBP5 [1:1000] (refs. [12,14] Sigma Prestige; #HPA047379; RRID:AB_10965845), anti-GTPBP10 [1:1000] (ref. [11] Novusbio; #NBP1-85055; RRID:AB_11037644), anti-bL32m [1:1000] (ref. [11] gift from Prof. P. Rehling; PRAB4957), anti-uS14m [1:1000] (Proteintech; #16301-1-AP; RRID:AB_2878240). Primary monoclonal anti-mouse antibody used in this study (dilution is indicated in brackets): anti-uL3 [1:500] (Proteintech; #66130-1-IG; RRID: AB_2881529).

**Cryo-EM sample preparation, data collection, and processing**. Purified mtLSU or 55S mitoribosome samples (4 µL) were applied to freshly glow discharged R 3.5/1 holey carbon grids (Quantifoil) that were precoated with a 2–3 nm carbon layer using a Leica EM ACE600 coater. Prior to flash freezing in liquid ethane, the samples were incubated on the grid for 30 s in a Vitrobot MarkIV (Thermo Fisher Scientific) at 4 °C and 100% humidity and subsequently blotted for 3 s seconds with a blot force of 0. Cryo-EM data collection was performed with SerialEM[37] using a Titan Krios transmission electron microscope (Thermo Fisher Scientific) operated at 300 keV. Images were acquired in EFTEM mode with a slit width of 20 eV using a GIF quantum energy filter and a K3 direct electron detector (Gatan) at a nominal magnification of 81,000× corresponding to a calibrated pixel size of 1.05 Å/pixel. Exposures were recorded in counting mode with a dose rate of ~20 $e^-$/px/s resulting in a total dose of 36–40 $e^-$/Å² (see Supplementary Table 1–3) that was fractionated into 40 movie frames. Motion correction, CTF-estimation, particle picking were

performed on the fly using Warp[38]. Particle extraction was performed with Relion[39] (dataset 1; 3-fold binned) or with Warp (dataset 2, dataset 3; unbinned).

For dataset 1 and dataset 2, particles were subjected to 2D classification followed by consensus 3D refinement using ab initio model created in cryoSPARC[40] as reference. Particles were then subjected to 3D classification without image alignment. In addition to the assembly factor-containing particles, both datasets contained two classes that resemble the mtLSU assembly intermediates previously observed in wild-type cells, which both contain the MALSU1-L0R8F8-mtACP module and differ in their rRNA folding state and the presence of bL36m[5,14].

The assembly factor-containing particles were unbinned by re-extraction (dataset 1) and subjected to 3D classification using a soft mask around the interfacial rRNA region where extra density for factors was visible (Supplementary Fig. 1, 3). Particle subsets were then subjected to CTF refinement (dataset 1 and 2) and Bayesian polishing (dataset 1). Further separation of particle populations with differing PTC conformations or subtle conformational differences within factors was achieved by using soft masks and a regularization parameter of T = 100 in Relion. Final maps were obtained by gold-standard 3D refinement followed by post-processing in Relion. Focused refinements using soft masks were used to obtain improved maps for conformationally flexible regions. For dataset 3, 2D classification and initial further steps were carried out in cryoSPARC (Supplementary Fig. 5). Good classes were selected and a subset of 150,000 particles was used to generate three ab initio models, which clearly resembled the full 55S mitoribosome, the mtLSU and the mtSSU. Particles were classified into these three classes by supervised classification in cryoSPARC and the mtLSU particle subset subject to consensus 3D refinement in Relion using the model from cryoSPARC as reference. Particles were further classified without image alignment, followed by focused classification with a soft mask around the interfacial rRNA and focused classification using a mask around GTPBP6 and T = 100. This led to three particle subsets containing GTPBP6. The first two represented different PTC conformations and were refined to high resolution, while the third resembled PTC conformation 1 but lacked clear density for h69. Final maps were obtained by gold-standard 3D refinement followed by post-processing in Relion. Local resolution was estimated using Relion. Figures were prepared with Chimera[41] and ChimeraX[42].

**Model building and refinement**. An initial model for the MTERF4-NSUN4 intermediate was obtained by rigid-body fitting the previously reported structure of a mtLSU assembly intermediate (PDB 5OOL)[5] and the previously reported crystal structure of the MTERF4-NSUN4 complex (PDB 4FZV)[16] into the density in Chimera. The model was manually adjusted and rebuilt in Coot[43] and served as a starting model for the remaining structures. The model of GTPBP5 was generated by docking a homology model generated with SwissModel[44] into the density followed by manual rebuilding in Coot. The GTPase domain of GTPBP5 showed poor density and only allowed for docking and adjusting of the homology mode and was therefore omitted from the final model. The final model contains residues 74–221 of GTPBP5. The model of GTPBP6 was generated by docking a homology model generated by SwissModel into the density followed by manual rebuilding in Coot. The final model of GTPBP6 comprises residues 94–515, but residues 106–118, 424–431 and the loop contacting h69 (106–118) are invisible in the mtLSU biogenesis intermediate containing GTPBP6. Models were built and interpreted using both the unsharpened and post-processed maps. Refinement was carried out using the unsharpened maps in phenix.real_space_refine[45] with restraints for 2'-methyl-UMP generated by phenix.elbow[46]. Model quality was assessed with MolProbity within the phenix suite[47]. Figures were generated with ChimeraX.

Notably, the reconstructions obtained from dataset 1 lack density for parts of uL24m, bL20m and mL42, while they show strong density in the reconstructions from dataset 2. In addition, we observed extra densities close to a number of surface-exposed cysteine residues in the reconstructions from dataset 1, accompanied by slight rearrangements of some protein and RNA loops. We speculate that these densities may represent covalent adducts to the sulfhydryl groups of cysteines, which may be a result of partial oxidation during sample preparation. Overall, we observed the following major differences between the MTERF4-NSUN4 and MTERF4-NSUN4-GTPBP5 intermediates from the first and the second dataset. First, the rRNA wrapping over MTERF4 is better resolved in dataset 2, showing that it forms a helical structure (Supplementary Fig. 4a). This is supported by basic residues in MTERF4 that form a charge-complementary binding groove, as well as two potential base-binding pockets (Supplementary Fig. 4b). Second, a part of the uL2m C-tail (residues 270-284) occupies a different path than in dataset 1 (Supplementary Fig. 4c). Third, the GTPase domain of GTPBP5 is better resolved in dataset 2, which allowed rigid-body fitting of a homology model (Supplementary Fig. 4d).

**Expression and purification of human GTPBP6**. Human Δ43GTPBP6 was cloned into the pET-derived vector 14-C (gift from Scott Gradia; Addgene plasmid #48309; http://n2t.net/addgene:48309; RRID:Addgene_48309) via ligation-independent cloning (primer sequences: Supplementary Table 4). 6xHis-MPB-tagged Δ43GTPBP6 was expressed in *E.coli* BL21 (DE3) RIL cells (Merck Millipore) grown in LB media. Cells were grown to an optical density at 600 nm of 0.5 at

37 °C and protein expression was subsequently induced with 0.15 mM isopropyl β-d-1-thiogalactopyranoside at 16 °C for 18 h. Cells were collected by centrifugation, resuspended in lysis buffer (300 mM NaCl, 50 mM Na-HEPES pH 7.4, 10% (v/v) glycerol, 30 mM imidazole pH 8.0, 2 mM DTT, 0.284 µg/ml leupeptin, 1.37 µg/ml pepstatin, 0.17 mg/ml PMSF, and 0.33 mg/ml benzamidine) and immediately used for protein purification performed at 4 °C. The cells were lysed by sonication and the lysate was cleared by centrifugation (87,200 × $g$, 4 °C, 30 min). The supernatant was applied to a HisTrap HP 5 ml column (GE Healthcare), preequilibrated in lysis buffer. The column was washed with 9.5 CV high-salt buffer (1000 mM NaCl, 50 mM Na-HEPES pH 7.4, 10% (v/v) glycerol, 30 mM imidazole pH 8.0, 2 mM DTT, 0.284 µg/ml leupeptin, 1.37 µg/ml pepstatin, 0.17 mg/ml PMSF, and 0.33 mg/ml benzamidine), and 9.5 CV low-salt buffer (150 mM NaCl, 50 mM Na-HEPES pH 7.4, 10% (v/v) glycerol, 30 mM imidazole pH 8.0 and 2 mM DTT). The sample was then eluted using nickel elution buffer (150 mM NaCl, 50 mM Na-HEPES pH 7.4, 10% (v/v) glycerol, 500 mM imidazole pH 8.0 and 2 mM DTT). The eluted protein was dialysed overnight in dialysis buffer (150 mM NaCl, 50 mM Na-HEPES pH 7.4, 15 mM imidazole pH 8.0, 10% (v/v) glycerol and 2 mM DTT) in the presence of 4 mg His-tagged TEV protease at 4 °C. The dialysed sample was applied to a HiTrap Heparin HP 5 ml column (GE Healthcare), preequilibrated in Buffer A (20 mM Na-HEPES pH 7.4, 10% (v/v) glycerol, 2 mM DTT) with 7.5 % Buffer B (2 M NaCl, 20 mM Na-HEPES pH 7.4, 10% (v/v) glycerol, 2 mM DTT). The protein was eluted with a linear salt gradient from 7.5–50% Buffer B and peak fractions containing GTPBP6 were collected and reapplied to a His Trap HP 5 ml column in the presence of 40 mM imidazole pH 8.0 to remove cleaved His-tagged MBP and TEV protease. The flow-through containing GTPBP6 was collected and applied to a Superdex 75 10/300 GL column (GE Healthcare) equilibrated in size exclusion buffer (70 mM NH$_4$Cl, 30 mM KCl, 7 mM MgCl$_2$, 20 mM TRIS-HCl pH 7.4, 10% (v/v) glycerol, 5 mM DTT). Peak fractions were assessed by SDS-PAGE and Coomassie staining and pooled. The protein was concentrated to ~65 mM using a MWCO 30,000 Amicon Ultra Centrifugal Filter (Merck), flash-frozen and stored at −80 °C until use.

**LC-MS/MS analysis**. Proteins were separated by polyacrylamide gel electrophoresis on a 4–12% gradient gel (NuPAGE, Invitrogen). After Coomassie staining, lanes were cut into 12 slices and proteins were reduced by dithiothreitol, alkylated by iodoacetamide, and digested with trypsin in-gel. Extracted peptides were vacuum-dried and subsequently resuspended in 2% acetonitrile (ACN, v/v)/0.05% trifluoroacetic acid (TFA, v/v). Peptides were measured on a QExactive HF Mass Spectrometer coupled to a Dionex UltiMate 3000 UHPLC system (both Thermo Fisher Scientific) equipped with an in house-packed C18 column (ReproSil-Pur 120 C18-AQ, 1.9 µm pore size, 75 µm inner diameter, 30 cm length, Dr. Maisch GmbH). Peptides were separated applying the following gradient: mobile phase A consisted of 0.1% formic acid (FA, v/v), mobile phase B of 80% ACN/0.08% FA (v/v). The gradient started at 5% B, increasing to 10% B within 3 min, followed by a continuous increase to 46% B within 45 min, and then keeping B constant at 90% for 8 min. After each gradient the column was again equilibrated to 5% B for 2 min. The flow rate was set to 300 nL/min. MS1 full scans were acquired with a resolution of 60,000, an injection time (IT) of 50 ms and an automatic gain control (AGC) target of 1 × 10$^6$. Dynamic exclusion (DE) was set to 30 s. MS2 spectra were acquired of the 30 most abundant precursor ions; the resolution was set to 15,000; the IT was set to 60 ms and the AGC target to 1 × 10$^5$. Fragmentation was enforced by higher-energy collisional dissociation (HCD) at 28% NCE. Acquired raw data were analyzed by MaxQuant[48] (v. 1.6.0.1) applying default settings and enabled 'match between runs' option. Proteins were quantified based on their iBAQ value.

The mass spectrometry proteomics data have been deposited to the ProteomeXchange Consortium via the PRIDE[49] partner repository with the dataset identifier PXD023502. Project Name: GTPase-driven maturation of the human mitoribosomal peptidyl transferase center; Project accession: PXD023502.

**Bisulfite sequencing to monitor 12S-m⁵C1488 and 12S-m⁴C1486**. To ensure that the accumulation of NSUN4 on the mtLSU in GTPBP6-deficient cells does not affect its second function as a methyltransferase modifying the 12S rRNA at position 1488, we analyzed the modification status of 12S-m⁵C1488 and, as a control, 12S-m⁴C1486, by subjecting DNase-treated total RNA from HEK293T wild-type and *Gtpbp6*⁻/⁻ cells to bisulfite sequencing[50,51]. Bisulfite treatment was performed using the EpiTect Bisulfite kit (Qiagen) according to the manufacturer's instructions. Deamination was performed by three cycles of incubation at 70 °C for 5 min and at 60 °C for 60 min. Samples were purified using mini Quick spin RNA columns (Roche) and the desulphonated in Tris pH 9.0 for 30 min at 37 °C. RNA was extracted using phenol:chloroform, precipitated and reverse transcribed from the 12S-m⁵C841_RT primer (Primer sequences: Supplementary Table 4) using Superscript III reverse transcriptase (Thermo) according to the manufacturer's instructions. A 70-nt fragment of the 12S rRNA was amplified by PCR (Primer sequences: Supplementary Table 4) and cloned using a TOPO-TA kit (Thermo). Clones were sequenced at Eurofins Genomics using the T7 primer and only sequences in which all cytosines (disregarding C1486 and C1488) were converted to uracil/thymine were used for the presented analysis (Supplementary Fig. 6c).

**Reporting summary**. Further information on research design is available in the Nature Research Reporting Summary linked to this article.

## Data availability

The electron density reconstructions and structure coordinates were deposited with the Electron Microscopy Database (EMDB) under accession codes EMD-12865, EMD-12872, EMD-12867, EMD-12868, EMD-12870, EMD-12869, EMD-12871, and with the Protein Data Bank (PDB) under accession codes 7OF0 [https://doi.org/10.2210/pdb7OF0/pdb], 7OF7 [https://doi.org/10.2210/pdb7OF7/pdb], 7OF2 [https://doi.org/10.2210/pdb7OF2/pdb], 7OF3 [https://doi.org/10.2210/pdb7OF3/pdb], 7OF5 [https://doi.org/10.2210/pdb7OF5/pdb], 7OF4 [https://doi.org/10.2210/pdb7OF4/pdb], and 7OF6 [https://doi.org/10.2210/pdb7OF6/pdb]. The mass spectrometry proteomics data have been deposited to the ProteomeXchange Consortium with the dataset identifier PXD023502. Material will be available upon reasonable request. Source data are provided with this paper.

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

## Acknowledgements

We thank Peter Rehling for providing antibodies. We thank Christian Dienemann and Ulrich Steuerwald for support with cryo-EM sample preparation and data acquisition. This work was funded by the Deutsche Forschungsgemeinschaft by the Emmy-Noether grant [RI 2715/1-1 to R.R.-D.]; the Excellence Cluster [EXC 2067/1-390729940 to R.R.-D. and H.S.H.]; the Forschergruppe 2848 [to H.S.H.], the Collaborative Research Center [SFB860 to R.R.-D., K.E.B., and H.U.; SFB1190 to H.S.H.]; and the Max Planck Society [to H.U.]. We acknowledge support by the Open Access Publication Funds of the Göttingen University.

## Author contributions

R.R.-D. and H.S.H. designed the study. H.S.H. prepared samples for cryo-EM, collected and processed cryo-EM data, built and interpreted structural models. E.L. and F.N. isolated ribosome complexes from human cells. E.H. established ribosome purification strategy. A.L. and H.U. performed mass spectrometry analysis. K.E.B. performed the bisulfite sequencing. H.S.H., E.L., and R.R.-D. prepared figures. H.S.H. and R.R.-D. wrote the manuscript.

## Funding

## Competing interests

The authors declare no competing interests.
