## [Peer Review File · Nature Communications]

REVIEWERS' COMMENTS

Reviewer #1 (Remarks to the Author):

Hillen and colleagues use cryo-EM to illuminate the poorly understood process of GTPase-mediated mt-LSU assembly and peptidyltransferase center (PTC) maturation. Using a HEK293 cell line deficient in GTPBP6, a late-stage assembly factor, they isolate native mt-LSU intermediates and show how six biogenesis factors (GTPBP5, the MTERF4:NSUN4 complex, and the MALSU1:LOR8F8:mt-ACP module) bind premature mt-LSU and contribute to the folding of mt-rRNA. They then reconstitute mt-LSU assembly *in vitro* by complementing the biogenesis intermediates with recombinant GTPBP6. They propose a sequential model where binding of GTPBP6 triggers the release of MTERF4:NSUN4 and GTPBP5 from the assembling mt-LSU and remodels mt-rRNA to form a nearly mature PTC. Finally, the authors explore the involvement of GTPBP6 in the dissociation of mitoribosomes. They demonstrate that GTPBP6 can bind the mature mt-LSU and rearrange elements that contribute to inter-subunit bridges, providing a putative mechanism for GTPBP6-mediated ribosome recycling.

This is a well-executed and well-written structural study, and no further experiments are necessary before publication. However, I would recommend the authors take advantage of the more generous word count of Nature Communications to expand the introduction and discussion. The introduction should cover what is already known about the assembly of the mammalian mt-LSU, including identification of assembly factors and the current status of our structural understanding. The discussion should try and link the study with the late stages of the assembly of the bacterial ribosome. Which factors and processes are conserved, and which are specific to mammalian mitochondria? The discussion should also address what is known about GTPBP7 and GTPBP10, and why they were not have been observed in the cryo-EM maps despite their presence in the mass spectrometry data.

I commend the authors for uploading their proteomics data to ProteomXchange but also encourage them to upload their micrographs to EMPIAR, as they will be beneficial to the community working on ribosome biogenesis and the development of new cryo-EM methods.

Minor

1. Culture volumes should be provided in the Methods.
2. Abbreviations should be defined in the figure legend. For example, "MRP" in Fig. 1.
3. In general, the figures are excellent but the pink of GTPBP6 and the red of the mt-rRNA were hard for me to distinguish in many of the figures where they overlap.
4. Please include an mt-rRNA secondary structure diagram to show how the rRNA/PTC changes during the maturation process.
5. Please check the .cif files associated with dataset 3. They display abnormally in Coot v.0.9.3.
6. Please carefully check the rRNA for all atomic models. For example, nucleotides 2910-2911 are incorrectly positioned in the dataset 1 maps.
7. Given the high resolution of the maps, is it possible to map all post-transcriptional and post-translational modifications that have taken place in the mt-LSU? Where modifications are present, they should be included in the atomic model.
8. Please check the positions of the modeled Mg ions. Not all have density, for example Mg 3345 in the MTERF4-NSUN4 model.
9. To aid structural comparison by users, it would help if all maps and models are aligned before deposition.

Reviewer #2 (Remarks to the Author):

In this manuscript Hillen et. al. present two novel human mitoribosome assembly intermediates

isolated from a GTPBP6 deficient cell line. GTPBP6 is a dual function enzyme with both ATPase and GTPase activities and its GTPase activity is required for mitoribosome biogenesis progression and ribosome splitting.

Deletion of GTPBP6 resulted in the accumulation of assembly intermediates containing the MTERF4-NSUN4 complex and GTPBP5, both involved in peptidyl transferase center (PTC) maturation. In addition, the authors show that further addition of GTPBP6 to isolated intermediates progresses biogenesis by folding the PTC to a near mature stage. Lastly, the authors propose a mechanism of GTPBP6 ribosome recycling by examining its binding site on mature mitoribosomes using cryoEM. In the opinion of this reviewer, this manuscript should be published in Nature Communications provided that the following issues are addressed.

Major points:

1. In the ribosome assembly field intermediates originating from cells in which a key assembly factor was depleted should be treated with caution and the authors should acknowledge that some of their observations may be due to the absence of GTPBP6.
2. Based on the local resolution analyses in the supplement, it is clear that the well-ordered solvent exposed region majorly contributes to the high overall resolution. To visualize the quality of the map around each of the assembly factors, the authors should prepare supplementary density figures highlighting local resolution in each area.
3. For figures in which high levels of detail are shown (Fig. 2, 3, 4c; supplementary figure 4g) corresponding density figures should be provided.
4. For their reconstitution experiment the authors used GTPBP6 together with particles obtained from a GTPBP6^{-/-} cell line. In particular both ATP and GTP were added, presumably to occupy what the authors previously characterized as ATPase and GTPase activities. However, as ATP is absent from any of the presented structures, the authors should elaborate on the rationale of this experiment as well as the obtained results.
5. As mentioned above (point 1), depletion experiments can accumulate "off-pathway" particles. In addition, to clarify the proposed model (extended data figure 5), the authors should indicate which of the intermediates are based on their obtained structures, and which correspond to models.
6. As three studies have now appeared on biorxiv that describe human large subunit assembly intermediates, the authors may want to consider their data in light of the other data to put their findings into a broader context and to unify an assembly pathway.

Minor points:

1. Can the authors explain their choice of the initial assembly intermediate (extended data fig. 5)? Without clarity on rRNA structure in the figure, the starting model seems closer to a previously visualised & later stage intermediate (Brown et. al 2017). Data obtained from Trypanosoma intermediates further suggests that the MALSU1 complex arrives with Mtg2 (GTPBP5).
2. Line 137: "sequential sequence" is a tautology.
3. For Figure 2a-c the comparison with the mature mtLSU could be helpful.
4. It is currently unclear which reconstructions (i.e. from which of the two datasets) were used for Figure 2.

Reviewer #3 (Remarks to the Author):

Many thanks for asking me to look at these two papers from the labs of Rorbach and Richter-Dennerlein. It was a pleasure. I'd really appreciate it if I could review the two of them together rather than try and dissect them or just repeat some of my comments for each of them. Further, I think it is

really useful to review them together and commend you on asking reviewers to look at both together. I have absolutely no expertise in structural biology but I think I can comment more generally. Both pieces of work were of very high quality and address the pathway of mitoribosome biogenesis in human cell lines, particularly the maturation of the large subunit. The approach was similar in that Cipullo isolates partial complexes in the GTPBP5 KO cell line and Hellen in the GTPBP6 KO line. In addition, Cipullo use a GTPBP5 tagged IP and Hellen use an in vitro approach of adding back purified GTPBP6 to their partial complexes. There is substantial similarity between what is found by cryoEM with the GTPBP5 IP and the GTPBP6 KO but there are differences. For instance, in the IP Cipullo finds GTPBP7, MRM2 and intriguingly mt-EF-Tu. These are all not present in the cryoEM from the GTPBP6 KO. Why is this ? Whilst intriguing, I am a little worried that Cipullo claim from these images that mtEF-Tu is involved in assembly of the mitoribosome. I think this is speculation and would need more supporting evidence from whole cell studies. There is strong evidence that GTPBP6 acts downstream of GTPBP5 and the in vitro studies, where purified GTPBP6 is added back and images of a more mature mtLSU with GTPBP6 bound and GTPBP5 + NSUN/MTERF4 absent does make one think that perhaps indeed GTPBP6 has somehow displaced this proteins. However, it is surely possible that GTPBP6 has bound to some intermediates in the preps that have lost these other components naturally and GTPBP6 has bound the free sites. I would be more convinced if a similar GTPBP6 IP had been performed from whole cells. However, it is entirely possible that GTPBP6 has indeed displaced these components.

Overall, I am impressed by the quality of both sets of data. There are some weaknesses and if the manuscripts can be toned down a little to indicate where weaknesses of their data interpretation may lie then I would be supportive of publication. One thing that intrigues me is that in the Cipullo paper there is no mention of GTPBP6. It is a ghost! Have Cipullo never come across this protein ? Surely they must at least speculate that on the basis of their images it is very likely that other assembly factors need to function further downstream of GTPBP5 to fully mature the LSU and that GTPBP6 could be one ? Then together these two manuscripts read very well and the GTPBP6 manuscript follows neatly on from the GTPBP5 work. Finally, I liked the final assembly figure of Cipullo and was not massively wowed by the video in Heller. Could I ask that a similar figure be included to help the reader in Heller et al. ?

Response to the reviewers

Reviewer #1 (Remarks to the Author):

Hillen and colleagues use cryo-EM to illuminate the poorly understood process of GTPase-mediated mt-LSU assembly and peptidyltransferase center (PTC) maturation. Using a HEK293 cell line deficient in GTPBP6, a late-stage assembly factor, they isolate native mt-LSU intermediates and show how six biogenesis factors (GTPBP5, the MTERF4:NSUN4 complex, and the MALSU1:LOR8F8:mt-ACP module) bind premature mt-LSU and contribute to the folding of mt-rRNA. They then reconstitute mt-LSU assembly in vitro by complementing the biogenesis intermediates with recombinant GTPBP6. They propose a sequential model where binding of GTPBP6 triggers the release of MTERF4:NSUN4 and GTPBP5 from the assembling mt-LSU and remodels mt-rRNA to form a nearly mature PTC. Finally, the authors explore the involvement of GTPBP6 in the dissociation of mitoribosomes. They demonstrate that GTPBP6 can bind the mature mt-LSU and rearrange elements that contribute to inter-subunit bridges, providing a putative mechanism for GTPBP6-mediated ribosome recycling.

This is a well-executed and well-written structural study, and no further experiments are necessary before publication. However, I would recommend the authors take advantage of the more generous word count of Nature Communications to expand the introduction and discussion. The introduction should cover what is already known about the assembly of the mammalian mt-LSU, including identification of assembly factors and the current status of our structural understanding. The discussion should try and link the study with the late stages of the assembly of the bacterial ribosome. Which factors and processes are conserved, and which are specific to mammalian mitochondria? The discussion should also address what is known about GTPBP7 and GTPBP10, and why they were not have been observed in the cryo-EM maps despite their presence in the mass spectrometry data.

We thank the reviewer for the kind words. We have extended the introduction and discussion as suggested.

I commend the authors for uploading their proteomics data to ProteomXchange but also encourage them to upload their micrographs to EMPIAR, as they will be beneficial to the community working on ribosome biogenesis and the development of new cryo-EM methods.

We agree with the reviewer and plan to do so after publication.

Minor

1. Culture volumes should be provided in the Methods.

We provided the information accordingly.

2. Abbreviations should be defined in the figure legend. For example, “MRP” in Fig. 1.

We defined abbreviations in the figure legends.

3. In general, the figures are excellent but the pink of GTPBP6 and the red of the mt-rRNA were hard for me to distinguish in many of the figures where they overlap.

To address this, we have modified Figure 2 and colored the three conserved loops of GTPBP5 in cyan throughout the figure to enhance contrast to the RNA. We have additionally simplified this figure by changing the depiction slightly, and omitting bases and residues not relevant to the discussion.

4. Please include an mt-rRNA secondary structure diagram to show how the rRNA/PTC changes during the maturation process.

We thank the reviewer for the suggestion and have included a simplified secondary structure diagram of the mt-rRNA in Figure 1. This shows the secondary structure of the mature mt-rRNA with the regions that undergo changes during the described maturation steps in red.

5. Please check the .cif files associated with dataset 3. They display abnormally in Coot v.0.9.3.

We thank the reviewer for pointing this out, the issue should be fixed now.

6. Please carefully check the rRNA for all atomic models. For example, nucleotides 2910-2911 are incorrectly positioned in the dataset 1 maps.

We have carefully re-evaluated all models and maps prior to final submission to the PDB.

7. Given the high resolution of the maps, is it possible to map all post-transcriptional and post-translational modifications that have taken place in the mt-LSU? Where modifications are present, they should be included in the atomic model.

We have modeled the described 2'-O-methylations of the 16S rRNA in all models except for those representing the MTERF4-NSUN4-bound state prior to GTPBP5 binding. In this state, the modified residues are partially mobile and thus have less clear density. As we do not know with certainty whether this state is prior or after methylation, we refrained from modeling these modification. To our knowledge, the only other known modifications are m¹A2617 and PsiU3067. We do not observe density for the former, and would likely not be able to distinguish the latter from unmodified U.

While the overall achieved resolution is comparably high and allows to distinguish between the presence or absence of known modifications in well-ordered regions, it is still very difficult to identify and model modifications that may be present but are not well characterized. Notably, we observed several instances of extra densities near RNA or protein residues that could potentially correspond to covalent modifications. In particular, the sample for dataset 1 may have undergone partial oxidation, as we observed densities near several solvent-exposed cysteine residues accompanied by slight rearrangements of protein backbone and RNA bases near these residues (see methods section). These extra densities likely correspond to covalent oxidation adducts, but we refrained from modeling these due to their unknown nature and lack of clear features.

8. Please check the positions of the modeled Mg ions. Not all have density, for example Mg 3345 in the MTERF4-NSUN4 model.

We have revised the Mg ions in all models.

9. To aid structural comparison by users, it would help if all maps and models are aligned before deposition.

We agree with the reviewer, and have thus superimposed all final maps and models that were submitted to the PDB.

Reviewer #2 (Remarks to the Author):

In this manuscript Hillen et. al. present two novel human mitoribosome assembly intermediates isolated from a GTPBP6 deficient cell line. GTPBP6 is a dual function enzyme with both ATPase and GTPase activities and its GTPase activity is required for mitoribosome biogenesis progression and ribosome splitting.

Deletion of GTPBP6 resulted in the accumulation of assembly intermediates containing the MTERF4-NSUN4 complex and GTPBP5, both involved in peptidyl transferase center (PTC) maturation. In addition, the authors show that further addition of GTPBP6 to isolated intermediates progresses biogenesis by folding the PTC to a near mature stage. Lastly, the authors propose a mechanism of GTPBP6 ribosome recycling by examining its binding site on mature mitoribosomes using cryoEM.

In the opinion of this reviewer, this manuscript should be published in Nature Communications provided that the following issues are addressed.

Major points:

1. In the ribosome assembly field intermediates originating from cells in which a key assembly factor was depleted should be treated with caution and the authors should acknowledge that some of their observations may be due to the absence of GTPBP6.

We agree with the reviewer and discussed this possibility in the revised manuscript.

2. Based on the local resolution analyses in the supplement, it is clear that the well-ordered solvent exposed region majorly contributes to the high overall resolution. To visualize the quality of the map around each of the assembly factors, the authors should prepare supplementary density figures highlighting local resolution in each area.

In the local resolution figures, blue coloring corresponds to high resolution while red corresponds to lower resolutions. Thus, the most rigid core, non-solvent exposed region displays highest resolution, while the solvent-exposed and more flexible regions display lower resolution, as is typical for single-particle cryo-EM maps. As suggested by the reviewer, we have added density figures for each of the factors we describe (Supplementary Fig. 2 and Supplementary Fig. 4)

3. For figures in which high levels of detail are shown (Fig. 2, 3, 4c; supplementary figure 4g) corresponding density figures should be provided.

We agree in principle with the reviewer. However, as the mentioned figures already contain a lot of information, we feel that adding density would disrupt the clarity and

the resulting figures would fail to convey the model-to-map fit appropriately. As an alternative, we have included close-up density figures for the PTC-interacting regions of GTPBP5 and GTPBP6 in Supplementary Fig. 2 and 4. We believe this can convey a good impression of the map quality in the regions relevant to our results without sacrificing clarity of our main figures, and hope the reviewer agrees with this assessment.

4. For their reconstitution experiment the authors used GTPBP6 together with particles obtained from a GTPBP6^{-/-} cell line. In particular both ATP and GTP were added, presumably to occupy what the authors previously characterized as ATPase and GTPase activities. However, as ATP is absent from any of the presented structures, the authors should elaborate on the rationale of this experiment as well as the obtained results.

We recently measured ribosome dissociation activity of GTPBP6 in a heterologous in vitro system (Lavdovskaia et al., 2020) showing that ribosome recycling mediated by GTPBP6 is dependent on GTP binding, but not on GTP hydrolysis. The addition of ATP or non-hydrolysable ATP analogs had no influence on this function. It has been suggested that the bacterial homolog HflX has ATP-dependent RNA-helicase activity to split heat-damaged ribosomes (Dey et al., 2018). Prior to the structural analyses we did not have any evidence whether ATP is required for the second function of GTPBP6 as a ribosome biogenesis factor. Thus, we collected cryo-EM datasets of the assembly intermediates purified from GTPBP6^{-/-} cells and treated with recombinant GTPBP6 in the presence of both ATP and a non-hydrolysable ATP analog. However, in neither case could we observe any bound nucleotide in the N-terminal domain. Thus, ATP is apparently also not required for its second function, and we decided to describe the sample for which we had the best cryo-EM dataset in this manuscript.

5. As mentioned above (point 1), depletion experiments can accumulate “off-pathway” particles. In addition, to clarify the proposed model (extended data figure 5), the authors should indicate which of the intermediates are based on their obtained structures, and which correspond to models.

We agree with the reviewer that this was not clear in our initial figure. We have decided to redesign the model figure and move it to the main figures (Figure 4). This new figure now includes clear indications of where the respective models are derived from.

6. As three studies have now appeared on biorxiv that describe human large subunit assembly intermediates, the authors may want to consider their data in light of the other data to put their findings into a broader context and to unify an assembly pathway.

We discussed our results in the context of the findings provided by Cipullo et al. (bioRxiv, doi.org/10.1101/2021.03.15.435084); Cheng et al. (bioRxiv, doi.org/10.1101/2021.03.17.435838); Lenarcic et al. (bioRxiv, doi.org/10.1101/2021.03.29.437532) and Chandrasekaran et al. (bioRxiv, doi.org/10.1101/2021.03.19.436169).

Minor points:

1. Can the authors explain their choice of the initial assembly intermediate (extended data fig. 5)? Without clarity on rRNA structure in the figure, the starting model seems closer to a previously visualised & later stage intermediate (Brown et. al 2017). Data obtained from Trypanosoma intermediates further suggests that the MALSU1 complex arrives with Mtg2 (GTPBP5).

Brown et al. (2017) described two assembly intermediates, one of which contains a highly unstructured interfacial rRNA (PDB 5OOM). We thus reasoned that this intermediate likely represents an earlier state than the ones observed in this study, and chose it as starting model. Note, however, that this model already contains the MALSU1 complex.

2. Line 137: “sequential sequence” is a tautology.

We have changed this accordingly.

3. For Figure 2a-c the comparison with the mature mtLSU could be helpful.

We thank the reviewer for the suggestion. We feel that including the comparison of the PTC to the mature state in this figure would disrupt the flow, because a further step is required (GTPBP6) which is depicted in the next figure. Instead, our revised model (Figure 4) contains close-up views of the PTC as it undergoes GTPase-mediated maturation, and this is also animated in the accompanying movie.

4. It is currently unclear which reconstructions (i.e. from which of the two datasets) were used for Figure 2.

Figure 2 was created using the models built from dataset 1, and we have now indicated this in the figure legend.

Reviewer #3 (Remarks to the Author):

Many thanks for asking me to look at these two papers from the labs of Rorbach and Richter-Dennerlein. It was a pleasure. I'd really appreciate it if I could review the two of them together rather than try and dissect them or just repeat some of my comments for each of them. Further, I think it is really useful to review them together and commend you on asking reviewers to look at both together. I have absolutely no expertise in structural biology but I think I can comment more generally. Both pieces of work were of very high quality and address the pathway of mitoribosome biogenesis in human cell lines, particularly the maturation of the large subunit. The approach was similar in that Cipullo isolates partial complexes in the GTPBP5 KO cell line and Hellen in the GTPBP6 KO line. In addition, Cipullo use a GTPBP5 tagged IP and Hellen use an in vitro approach of adding back purified GTPBP6 to their partial complexes. There is substantial similarity between what is found by cryoEM with the GTPBP5 IP and the GTPBP6 KO but there are differences. For instance, in the IP Cipullo finds GTPBP7, MRM2 and intriguingly mt-EF-Tu. These are all not present in the cryoEM from the GTPBP6 KO. Why is this ?

We can of course only speculate on the reason for the absence of these factors in our reconstructions. We did observe GTPBP7 in our mass spectrometry results, however, despite extensive classification efforts, we did not observe a particle

population showing density corresponding to GTPBP7. Based on the recent studies by the Ban group (Lenarcic et al., bioRxiv, doi.org/10.1101/2021.03.29.437532), by Cipullo et al. and Chandrasekaran et al. that appeared simultaneously with our work, we speculate that GTPBP7-binding may be flexible and highly transient. This could explain the absence of structurally interpretable density for GTPBP7 in our datasets, as the population of GTPBP7-bound mtLSU may be very small in our datasets. One important difference between our approaches is obviously that purification via GTPBP5-FLAG enriches for intermediates with this factor bound. In the GTPBP5-deficient cells, GTPBP5-bound mtLSU presents just a small subclass of particles. Thus, it is not unexpected that factors that may bind transiently together with GTPBP5, such as MRM2, are not dominant in our dataset. The absence of MRM2 in MS and in the structure also matches with our results that we observe corresponding density for 2'-O-methylation at position U3039, which is in contrast to MRM2-bound mtLSU provided by Cipullo et al. and clearly indicates that our GTPBP5-bound state represents an intermediate downstream of MRM2-action. This suggests that MRM2 can dissociate after methylating U3039, whereas GTPBP5 can remain bound until GTPBP6 binds.

Whilst intriguing, I am a little worried that Cipullo claim from these images that mtEF-Tu is involved in assembly of the mitoribosome. I think this is speculation and would need more supporting evidence from whole cell studies. There is strong evidence that GTPBP6 acts downstream of GTPBP5 and the in vitro studies, where purified GTPBP6 is added back and images of a more mature mtLSU with GTPBP6 bound and GTPBP5 + NSUN/MTERF4 absent does make one think that perhaps indeed GTPBP6 has somehow displaced these proteins. However, it is surely possible that GTPBP6 has bound to some intermediates in the preps that have lost these other components naturally and GTPBP6 has bound the free sites. I would be more convinced if a similar GTPBP6 IP had been performed from whole cells. However, it is entirely possible that GTPBP6 has indeed displaced these components.

We thank the reviewer for his/her comment and agree that it is also possible that GTPBP6 binds to mtLSU intermediates with free sites. Unfortunately, a similar IP for GTPBP6 as performed for GTPBP5 would not yield adequate amounts of mtLSU complexes. The interaction of GTPBP6 with the mitochondrial ribosome seems to be rather transient (Lavdovskaia et al., 2020). In addition, we have to be very careful with the expression levels of GTPBP6 as elevated levels will lead to ribosome splitting and translation deficiency.

Although we do not have direct evidence that GTPBP6 releases GTPBP5 and MTERF4-NSUN4, it is clear that the binding of GTPBP6 and MTERF4-NSUN4-GTPBP5 is mutually exclusive. In addition, the number of particles containing MTERF4-NSUN4-GTPBP5 was reduced by approximately 50% comparing dataset 2 with dataset 1, which also supports the hypothesis that GTPBP6 replaces these assembly factors. We have, however, reflected this uncertainty in our conclusions accordingly.

Overall, I am impressed by the quality of both sets of data. There are some weaknesses and if the manuscripts can be toned down a little to indicate where weaknesses of their data interpretation may lie then I would be supportive of

publication. One thing that intrigues me is that in the Cipullo paper there is no mention of GTPBP6. It is a ghost! Have Cipullo never come across this protein ? Surely they must at least speculate that on the basis of their images it is very likely that other assembly factors need to function further downstream of GTPBP5 to fully mature the LSU and that GTPBP6 could be one ? Then together these two manuscripts read very well and the GTPBP6 manuscript follows neatly on from the GTPBP5 work. Finally, I liked the final assembly figure of Cipullo and was not massively wowed by the video in Heller. Could I ask that a similar figure be included to help the reader in Heller et al. ?

We thank the reviewer for his/her suggestion and included a new main figure (Fig. 4), which contains both a simplified cartoon representation of the model as well as more detailed views of the PTC rearrangements that occur.